# Reconstruction for Defects of Total Nail Bed and Germinal Matrix Loss with Acellular Dermal Matrix Coverage and Subsequently Skin Graft

**DOI:** 10.3390/medicina56010017

**Published:** 2020-01-03

**Authors:** Tsung-Hsien Liu, Meng-Chien Hsieh, Ping-Ruey Chou, Shu-Hung Huang

**Affiliations:** 1Division of Plastic Surgery, Department of Surgery, Kaohsiung Medical University Hospital, Kaohsiung 807, Taiwan; aapr08l@gmail.com (T.-H.L.); williehsieh@hotmail.com (M.-C.H.); 2School of Medicine, College of Medicine, Kaohsiung Medical University, Kaohsiung 807, Taiwan; u105025047@gap.kmu.edu.tw; 3Department of Surgery, School of Medicine, College of Medicine, Kaohsiung Medical University, Kaohsiung 807, Taiwan; 4Regeneration Medicine and Cell Therapy Research Center, Kaohsiung Medical University, Kaohsiung 807, Taiwan

**Keywords:** nail bed loss, acellular dermal matrix, skin graft

## Abstract

*Background and Objectives:* Nail bed and germinal matrix loss due to wide excision for fingertip tumors or malignancy are occasionally encountered complications. These defects also result from severely comminuted fingertip crush injuries. Large-area dorsal finger or toenail bed defects, which usually present with phalangeal bone exposure, remain challenging regardless of the usage of different reconstruction strategies. This study aimed to evaluate the clinical outcome of a staged operation with an acellular dermal matrix coverage and subsequent skin graft as reconstruction for defects of total nail bed, germinal matrix loss, and bone exposure. *Materials and Methods:* From April 2018 to October 2019, four patients with total nail bed, germinal matrix, and bone exposure loss after surgery were enrolled in our series. A staged operation of the acellular dermal matrix coverage with subsequent skin graft was performed on these patients. Skin graft take rate, oncological prognosis, and cosmetic outcome were evaluated. Patients were followed up for 5–13 months. An excellent skin graft take rate with a satisfying aesthetic result without local malignancy recurrence was noted. Minimal functional deficit and donor site morbidity were reported. *Results:* A staged operation with acellular dermal matrix coverage and subsequent skin graft proves to serve as a feasible strategy for patients who experience total nail bed, germinal matrix loss, and bone exposure after surgery. *Conclusions:* This reconstruction method provides a reliable repair result, satisfying aesthetic outcomes, as well as having minimal functional deficits and donor site morbidity.

## 1. Introduction

Defects that result from total nail bed and germinal matrix removal are occasionally encountered in patients with digit malignancy who had undergone local wide excision. This type of defect is also seen in patients with severe crush and comminuted nail bed injuries. Several recent papers proposed the application of a more conservative, tissue-sparing local wide excision or Mohs micrographic surgery as they provide a similar long-term oncological outcome with preservation of digit function and a more acceptable body contour when compared to digital amputations [1,2,3,4,5]. However, local wide excision for digital malignancy often results in total nail bed and germinal matrix loss with large area of distal phalanx exposure. For bone exposure fingertip defects, several reconstruction options have been discussed in the literature, including skin grafts, composite grafts, bone shortening with local advancement flap, V-Y flap, reverse digital artery flap, adipofascial flap, cross-finger fascial flap, and free vascularized toe transfer [6,7,8,9,10,11,12,13]. These reconstruction strategies are usually accompanied by considerable donor site morbidity and postoperative body contour changes. To focus more on total nail bed loss of surgical removal or avulsion nail bed injury, the challenge for reconstruction lies in how the defects are located on the dorsal aspect where there is a lack of soft tissue and that can often be accompanied by large areas of phalangeal bone exposure. Fiedler et al. had reported the use of an acellular dermal matrix for nail bed avulsion or crush injuries with excellent results [14]. In our series, we applied acellular dermal matrix coverage for three weeks until the defects revealed full granulation tissue formation with subsequent full-thickness skin graft for patients that sustained total nail bed and germinal matrix removal. In this study, the demographic data, surgical details, and clinical results were reviewed to evaluate the clinical outcome of a staged operation with acellular dermal matrix coverage and subsequent skin graft for reconstruction of defects of total nail bed and germinal matrix loss with bone exposure.

## 2. Materials and Methods

### 2.1. Patients

From April 2018 to October 2019, four patients with total nail bed and germinal matrix removal with phalangeal bone exposure after surgical excision for fingertip tumor and malignancy were enrolled at Kaohsiung Medical University’s Chung-Ho Memorial Hospital. Three of them were male and one was female. The mean age was 57.3 years, ranging from 31 to 77 years old. Among these patients, three patients had lesions located on the toes and the other on the left ring finger. Two patients had a pathological diagnosis of melanoma from a previous biopsy. One patient was diagnosed with squamous cell carcinoma. The last patient that was impressed presented with melanoma on the right big toe for which he received a direct local wide excision without an initial biopsy (Table 1). Medical charts, past history, and biopsy results were reviewed. This study was approved by the Institutional Review Board of Kaohsiung Medical University’s Chung-Ho Memorial Hospital (KMUHIRB-E(I)-20190135).

### 2.2. Surgical Procedures

Our proposed procedure consists of two stages. In the first stage of surgery, we performed a local wide excision that included a total nail bed and germinal matrix removal for the eradication of digit malignancies under general anesthesia. Once the wide excision was completed with adequate margin control confirmed by intraoperative frozen sections, the defects usually included a large area of distal phalangeal bone exposure. We applied an acellular dermal matrix (PELNAC®, Gunze Corp., Osaka, Japan) to cover the defect as the first stage of the reconstruction. The acellular dermal matrix was fixed to the wound bed with non-absorbable sutures and dressed with silicone-foam dressings. The patients were discharged on the day of surgery. Ten days after, the acellular dermal matrix was removed and re-dressed with a new one at the outpatient clinic. Three patients received the second stage of operation at 21 days after the first stage and the fourth patient on day 25. During the second stage, the remnant acellular dermal matrix was removed. All cases were observed to have a healthy and well-nourished granulated wound bed. Full-thickness skin grafts harvested from the groin area were designed after accurate measurement of the wound bed.

### 2.3. Clinical Results

The size of the skin grafts ranged from 3 to 13.5 cm^2^, with a mean of 7.87 cm^2^ (Table 1). The harvested skin graft was bolstered with a tie-over dressing. Skin graft take rate was evaluated at day 14 after skin graft surgery, which revealed 100% in Case 1–3 and 98% in Case 4. In the period of follow-up, there was no evidence of recurrent skin cancer reported. These patients experienced minimal change in body contour, mild but acceptable functional deficit, and satisfying aesthetic results.

## 3. Case Illustrations

### 3.1. Case 1

A 64-year-old male sustained a wood splinter injury on his left ring fingertip years ago with subsequent chronic ulceration and intermittent tenderness. He first visited a dermatological clinic where a biopsy disclosed squamous cell carcinoma. He was referred to our department and received a local wide excision for malignancy eradication. In the first stage of the surgery, the total nail bed and germinal matrix were removed as part of the wide excision (Figure 1A). The intraoperative frozen section revealed adequate margin control. The defect then presented with a large area of distal phalangeal bone exposure (Figure 1B). An acellular dermal matrix was placed directly over the bony exposure defect and fixed with non-absorbable sutures. The acellular dermal matrix was re-dressed with a new piece at the outpatient department on postoperative day 10. Then, on day 21, a second operation was arranged. A well-granulated wound bed was noted after the removal of the remnant acellular dermal matrix (Figure 2A). A full-thickness skin graft measuring 2 × 1.5 cm^2^ was harvested from the groin area and grafted to the well-granulated wound bed (Figure 2B). Further follow up revealed excellent skin take with a satisfying cosmetic result (Figure 3A,B).

### 3.2. Case 2

A 77-year-old male sustained black discoloration over his left toenail for four years. Due to progressive enlargement of the discoloration, he went to the dermatology department where a biopsy revealed melanoma in situ. He was referred to our department and received the operation of local wide excision with total nailbed and germinal matrix removal. A large region of phalangeal bone exposure, measured at 3 × 4.5 cm^2^, were noted (Figure 4A). An acellular dermal matrix was applied onto the defect (Figure 4B). Three weeks later, healthy granulation growing over the phalangeal bone was noted (Figure 4C). He underwent the second-stage operation with the full-thickness skin graft harvested from the groin area. Post-operation day 42 photo revealed excellent skin graft take (Figure 4D). Intact flexion and extension functions of his big toe were also reported (Figure 4E,F).

## 4. Discussion

In our series, combined acellular dermal matrix coverage with subsequent skin graft repair provide excellent skin take results with acceptable cosmetic outcome for defects of total nail bed and germinal matrix removal. There were minimal donor site morbidity, negligible functional deficits, and limited change in body contour.

Many reconstructive options can be chosen for fingertip injuries with bone exposure. Martin-Playa et al. [6] and Tang et al. [10] had reviewed the use of skin graft, composite graft, adipofascial flap, local advancement flap, V-Y flap, first dorsal metacarpal artery flap, and reverse digital artery flap for fingertip defects. For large defects, free tissue transfer with a vascularized toe pulp or partial toe transfer may be selected [10]. Unfortunately, these methods are usually accompanied with considerable donor site morbidity, conspicuous distortion of the postoperative body contour, and even inconvenient functional deficits induced by elongated scarring.

For large nail bed defects with a preserved germinal matrix, Yang et al. [9] reported their experience in using a cross finger fascial flap combined with a split-thickness toe nail bed graft. Hsieh et al. [15] used a split-thickness toenail bed graft for avulsed nailbed defects. Fiedler et al. [14] recommended the use of a single-layer bovine acellular dermal matrix for the reconstruction of severe nail bed crush injuries. Nail bed repair is warranted for further nail plate growth and the prevention of an irregular and nonadherent nail plate.

Few literatures focus on patients with total defect of the nail bed and germinal matrix loss following surgical removal or comminuted crush injuries. The reconstruction challenges include large area of distal phalangeal bone exposure and a lack of soft tissue for local flap repair. However, as no remnant germinal matrix is preserved, nail plate growth is no longer possible. The major challenge remains in the reconstruction of a large defect of bone exposure. A dorsal reverse adipofascial flap might be a considerable reconstructive option [8,12,13]. Yet, the germinal matrix was sacrificed in our cases, which could inadvertently obstruct the origin of vascular supply for the dorsal reverse adipofascial flap [16]. As the safety distance for perforator preservation can be violated and compromised, the application of a dorsal reverse adipofascial flap in this circumstance should be reconsidered with extreme caution.

In this study, we used a staged strategy to reconstruct defects with total nail bed and germinal matrix loss. The most important benefit of a staged operation was the use of an acellular dermal matrix which can successfully convert a sizeable defect of bone exposure defect to a well-nourished granulation-covered wound bed. The acellular dermal matrix functions as a scaffold and promote granulation tissue ingrowth. In our experience, even for cases with an entire dorsal surface of distal phalangeal bone exposure, three weeks had been sufficient for the growth of granulation tissue. Once a new granulation-filled wound bed was observed, a second-stage operation of skin grafting was performed. The take rate of skin grafting in our series was 100%. Through the application of an acellular dermal matrix, we were able to step down on the reconstruction ladder. In consideration of donor site morbidity, full-thickness skin grafts harvested from the groin area are still considerably less than that of local flap or free tissue transfer. The ingrowth granulation tissue also worked as the replacement volume for total nail bed excision. The initial grossly depressed appearance would restore the lost soft tissue volume. This effect makes the long-term cosmetic outcome much more acceptable. The other advantage of our method is that both flexion and extension functions are not compromised.

Nevertheless, limitations of this article should not be ignored. First, the patients in our series sustained the defects from an oncologic excision. Thus, application of this method to complex crush trauma patients may not be suitable. Second, staged surgeries translate into longer hospital stays and a higher incurred financial burden. Third, the number of patients in our series is not sizeable with only a limited follow-up period. Further investigative studies are required to confirm our current results.

## 5. Conclusions

A staged operation with acellular dermal matrix coverage and subsequent skin graft repair for defects with total nail bed and germinal matrix loss is a feasible reconstructive method. This strategy provides reliable repair results, satisfying aesthetic outcomes, as well as having minimal functional deficits and donor site morbidity.

## Figures and Tables

**Figure 1 medicina-56-00017-f001:**
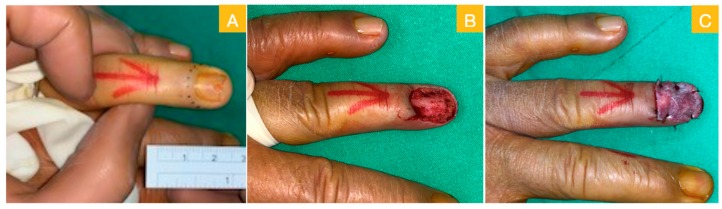
(**A**) Local wide excision with total nail bed and germinal matrix removal; (**B**) phalangeal bone exposure after wide excision; and (**C**) artificial dermis graft for temporary repair.

**Figure 2 medicina-56-00017-f002:**
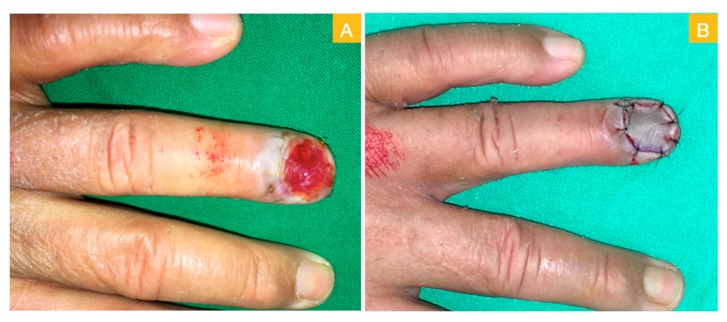
(**A**) Healthy granulation after artificial dermis graft and (**B**) repair with full-thickness skin graft.

**Figure 3 medicina-56-00017-f003:**
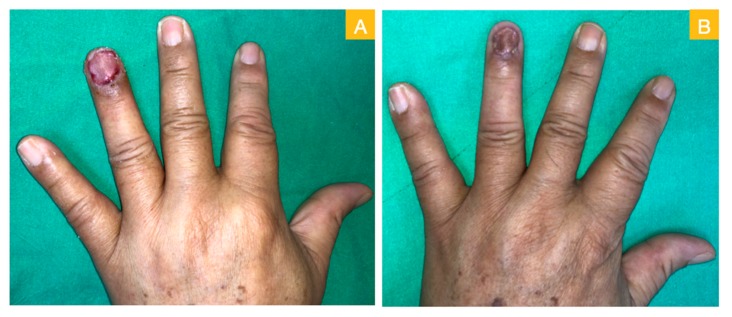
(**A**) Post-skin graft on day 11 and (**B**) post-skin graft on day 34.

**Figure 4 medicina-56-00017-f004:**
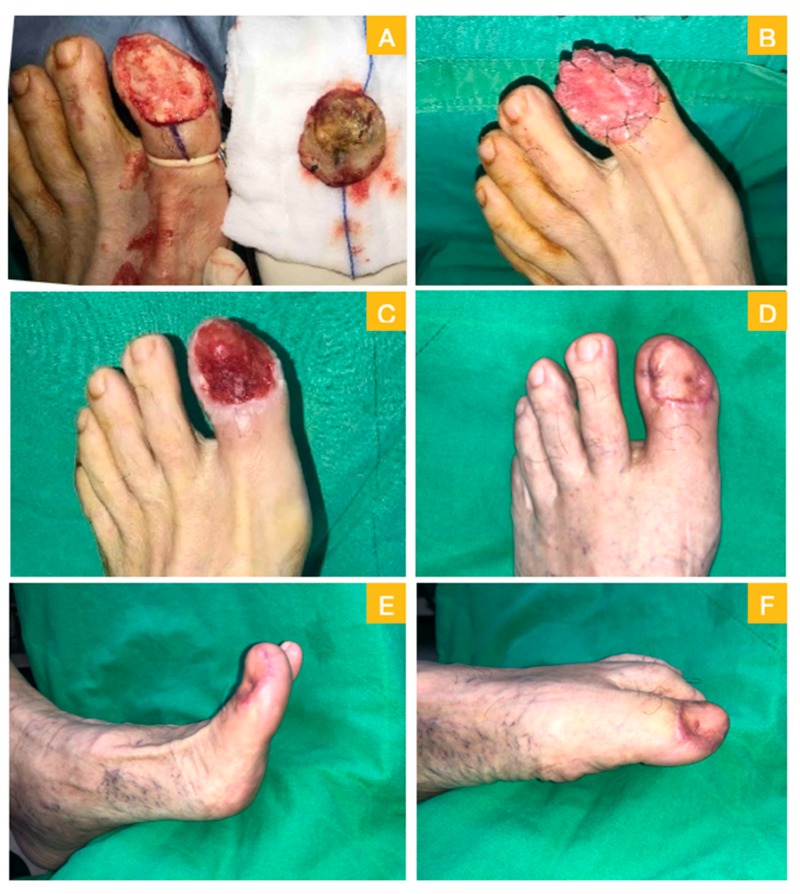
Case 2. (**A**) Wide excision with total nail bed and germinal matrix removal, bone exposure was noted; (**B**) acellular dermal matrix for stage one repair; (**C**) day 21 with healthy granulation; (**D**) 42 days after skin graft; and (**E,F**) intact toes movement.

**Table 1 medicina-56-00017-t001:** Patient demographics.

Case	Age	Sex	Past History	Location	Biopsy Report	Pathology With TNM Stage	Stage 1 Surgery	Stage 2 Surgery	Operation Interval	FTSG Take Rate	Follow up Period
1	77	M	HTN, prostate adenocarcinoma	left big toe	melanoma in situ	no residual tumor (pTisN0)	wide excision + PELNAC®	FTSG (3 × 3 cm^2^)	21 days	100%	13 months
2	57	F	HTN, HBV, breast fibroadenoma	right big toe	malignant melanoma	lentigo maligna (pTisN0)	wide excision + PELNAC®	FTSG (3 × 4.5cm^2^)	21 days	100%	10 months
3	64	M	HTN, dyslipidemia	left ring finger	SCC	no residual tumor (cT2N0M0 stage 2)	wide excision + PELNAC®	FTSG (2 × 1.5 cm^2^)	21 days	100%	9 months
4	31	M	nil	right big toe	nil, clinical suspected melanoma	hyperpigmentation in basal layer of epidermis	wide excision + PELNAC®	FTSG (3 × 2 cm^2^)	25 days	98%	5 months

HTN, hypertension; HBV, hepatitis B virus; SCC, squamous cell carcinoma; FTSG, full thickness skin graft.

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
