# Peer review of "Reconstruction for Defects of Total Nail Bed and Germinal Matrix Loss with Acellular Dermal Matrix Coverage and Subsequently Skin Graft"

_medicina, 2020, doi:10.3390/medicina56010017_

Round 1

Reviewer 1 Report

I recommend the following edits:

Line 37: change "defects" to defect

Line 38: change "literatures" to papers

Line 40: change "functions" to function

Line 41: change "result" to results

Line 42: delete the word "wound" and leave the word "exposure" as the last word in the sentence.

Line 43: replace the words "used in literatures" with discussed in the literature,

Line 51: change "result" to results

Line 51-52: replace the words "till well granulation formation" with (until defects revealed full granulation tissue formation) and place (  ) around the words "for defects resulting from total nail bed and germinal matrix removal"

Line 56: line should read "defects of total nail bed and germinal matrix loss, with bone exposure

Line 65: delete word "impressed" and replace with "presented"

Line 71: delete the words "of operations"

Line 75:  delete "presented with" and replace with "included a"

Line 89: add a comma after words "contour" and "deficit"

Line 90: change "result" to "results"

Line 164: add the word "more" after the word "much"

Author Response

Dear Reviewer:

Thank you very much for your comments on our manuscript. We have carefully considered the comments of the reviewers and revised our manuscript accordingly. The revised parts have been highlighted in the manuscript. Our responses to the reviewers' comments are as follows.

Major Criticisms:

Line 37: change "defects" to defect

ANSWER:

  Thank you for Reviewer’s comment, we agree with your points. We have revised it in introduction part (page 1, Line 36).

Line 38: change "literatures" to papers

ANSWER:

  Thank you for Reviewer’s comment, we agree with your points. We have revised it in introduction part (page 1, Line 37).

 Line 40: change "functions" to function

ANSWER:

  Thank you for Reviewer’s comment, we agree with your points. We have revised it in introduction part (page 1, Line 40).

Line 41: change "result" to results

ANSWER:

  Thank you for Reviewer’s comment, we agree with your points. We have revised it in introduction part (page 1, Line 41).

Line 42: delete the word "wound" and leave the word "exposure" as the last word in the sentence.

ANSWER:

  Thank you for Reviewer’s comment, we agree with your points. We have revised it in introduction part (page 1, Line 42).

Line 43: replace the words "used in literatures" with discussed in the literature,

ANSWER:

  Thank you for Reviewer’s comment, we agree with your points. We have revised it in introduction part (page 1, Line 43).

Line 51: change "result" to results

ANSWER:

  Thank you for Reviewer’s comment, we agree with your points. We have revised it in introduction part (page 2, Line 51).

Line 51-52: replace the words "till well granulation formation" with (until defects revealed full granulation tissue formation) and place (  ) around the words "for defects resulting from total nail bed and germinal matrix removal"

ANSWER:

  Thank you for Reviewer’s comment, we agree with your points. We have revised it in introduction part (page 2, Line 52-54).

Line 56: line should read "defects of total nail bed and germinal matrix loss, with bone exposure

ANSWER:

  Thank you for Reviewer’s comment, we agree with your points. We have revised it in introduction part (page 2, Line 57).

Line 65: delete word "impressed" and replace with "presented"

ANSWER:

  Thank you for Reviewer’s comment, we agree with your points. We have revised it in methods part (page 2, Line 66).

Line 71: delete the words "of operations"

ANSWER:

  Thank you for Reviewer’s comment, we agree with your points. We have revised it in methods part (page 2, Line 72).

Line 75:  delete "presented with" and replace with "included a"

ANSWER:

  Thank you for Reviewer’s comment, we agree with your points. We have revised it in methods part (page 2, Line 76).

Line 89: add a comma after words "contour" and "deficit"

ANSWER:

  Thank you for Reviewer’s comment, we agree with your points. We have revised it in methods part (page 2, Line 91).

Line 90: change "result" to "results"

ANSWER:

  Thank you for Reviewer’s comment, we agree with your points. We have revised it in methods part (page 2, Line 91).

Line 164: add the word "more" after the word "much"  

ANSWER:

  Thank you for Reviewer’s comment, we agree with your points. We have revised it in discussion part (page 6, Line 170).

Reviewer 2 Report

I think that the topic is of interest, but the scientific impact is more than limited.

It is a presentation of a small case series and not a real study. Due to this the scientific and clinical impact is limited. It is not real new that you can use a acellular dermal matrix together with a skin graft to cover these kind of defects. 

Author Response

Dear Reviewer:

Thank you very much for your comments on our manuscript. We have carefully considered the comments of the reviewers and revised our manuscript accordingly. The revised parts have been highlighted in the manuscript. Our responses to the reviewers' comments are as follows.

Major Criticisms:

I think that the topic is of interest, but the scientific impact is more than limited. It is a presentation of a small case series and not a real study. Due to this the scientific and clinical impact is limited. It is not real new that you can use a acellular dermal matrix together with a skin graft to cover these kind of defects. 

  ANSWER: 

  Thank you for precious Reviewer’s comment. We will take your suggestions into our further research considerations.

Reviewer 3 Report

This article is interesting. The English is good, the pictures are great.

I suggest to add these papers to the References:

1: Sisti A, Oliver JD, Nisi G. Fenestrated adipofascial reverse flap for the
reconstruction of fingertip amputations. Microsurgery. 2019 Sep;39(6):575. doi: 10.1002/micr.30480. Epub 2019 Jul 3. PubMed PMID: 31268559.

2: Idone F, Sisti A, Tassinari J, Nisi G. Cooling Composite Graft for Distal
Finger Amputation: A Reliable Alternative to Microsurgery Implantation. In Vivo. 2016 Jul-Aug;30(4):501-5. PubMed PMID: 27381615.

In the Discussion Section, please discuss more extensively this paper:

Idone F, Sisti A, Tassinari J, Nisi G. Fenestrated Adipofascial Reverse Flap: A Modified Technique for the Reconstruction of Fingertip Amputations. J Invest Surg. 2017;30:353-8.

Author Response

Dear Reviewer:

Thank you very much for your comments on our manuscript. We have carefully considered the comments of the reviewers and revised our manuscript accordingly. The revised parts have been highlighted in the manuscript. Our responses to the reviewers' comments are as follows.

Major Criticisms:

1. I suggest to add these papers to the References:

Sisti A, Oliver JD, Nisi G. Fenestrated adipofascial reverse flap for the
reconstruction of fingertip amputations. Microsurgery. 2019 Sep;39(6):575. doi: 10.1002/micr.30480. Epub 2019 Jul 3. PubMed PMID: 31268559.

Idone F, Sisti A, Tassinari J, Nisi G. Cooling Composite Graft for Distal
Finger Amputation: A Reliable Alternative to Microsurgery Implantation. In Vivo. 2016 Jul-Aug;30(4):501-5. PubMed PMID: 27381615.

ANSWER:

  Thank you for Reviewer’s suggestion, we agree with your points. We have added it in introduction part (page 2, Line 45), discussion part (page 153), and reference part (page 7, Line 212, 216).

2. In the Discussion Section, please discuss more extensively this paper:

Idone F, Sisti A, Tassinari J, Nisi G. Fenestrated Adipofascial Reverse Flap: A Modified Technique for the Reconstruction of Fingertip Amputations. J Invest Surg. 2017;30:353-8.

ANSWER:

  Thank you for Reviewer’s suggestion, we agree with your points. We have revised it in discussion part (page 6, Line 152-157).

Round 2

Reviewer 2 Report

I think that the impact is really limited!!